# Materials tactile logic via innervated soft thermochromic elastomers

Yang Jin[1,2], Yiliang Lin[1], Abolfazl Kiani[1,3], Ishan D. Joshipura [1], Mingqiao Ge[2] & Michael D. Dickey [1]

Conventional machines rely on rigid, centralized electronic components to make decisions, which limits complexity and scaling. Here, we show that decision making can be realized on the material-level without relying on semiconductor-based logic. Inspired by the distributed decision making that exists in the arms of an octopus, we present a completely soft, stretchable silicone composite doped with thermochromic pigments and innervated with liquid metal. The ability to deform the liquid metal couples geometric changes to Joule heating, thus enabling tunable thermo-mechanochromic sensing of touch and strain. In more complex circuits, deformation of the metal can redistribute electrical energy to distal portions of the network in a way that converts analog tactile 'inputs' into digital colorimetric 'outputs'. Using the material itself as the active player in the decision making process offers possibilities for creating entirely soft devices that respond locally to environmental interactions or act as embedded sensors for feedback loops.

[1] Department of Chemical and Biomolecular Engineering, North Carolina State University, 911 Partners Way, Raleigh, NC 27695, USA. [2] College of Textile and Clothing, Jiangnan University, 1800 Lihu Ave, Wuxi, Jiangsu Province 214122, China. [3] Department of Chemistry, University of Isfahan, Isfahan 8174673441, Iran. Correspondence and requests for materials should be addressed to M.G. (email: ge_mingqiao@126.com) or to M.D.D. (email: mddickey@ncsu.edu)

Nature abounds with soft, functional materials that, to date, have few synthetic analogs. For example, eyes (adaptive optics), skin (multimodal sensing and self-healing), nerve networks, and brains (computation) are built entirely from soft materials. Other entirely soft organisms, such as cephalopods, can change color and have nervous systems capable of localized responses to sensory inputs without the need for centralized processing (i.e., the brain). There is great interest in trying to mimic these functions synthetically using soft materials. Applications include soft robotics (for human assistance, prosthetics, and disaster response), human–computer interfaces, passive displays, adaptive surfaces for camouflage, and electronic skins.

This paper demonstrates that silicones innervated with liquid metal circuits—which are completely soft and stretchable—are capable of a variety of interesting functions based on the ability to direct (or locally redirect) electrical energy through these dynamic circuits. As an output, we use the energy to bring about color change due to the importance of visual cues, although in principle the energy could be used for other applications. These soft systems are capable of a variety of interesting bio-mimetic capabilities motivated herein, but perhaps the most interesting is the ability to perform "soft tactile distributed logic" in which the soft materials are themselves the active player in decision-making.

To understand this concept, consider that many physical systems—both natural and artificial—utilize sensors to gather information that must be processed to decide a response (so-called "sense-decide-respond" control loops)[1]. In humans, for example, nerve networks send sensory information to the brain, which processes the information to determine a response (muscle contraction). Likewise, machines (e.g., robots) utilize arrays of sensors that connect to a central computer, which processes the information before calculating a response. Both cases involve a central processor and complex "wiring" to connect all the components that limits scaling. There are exceptions. In humans, for example, reflexes bypass the brain to achieve a rapid, albeit less sophisticated response. In the octopus, neural ganglia process sensory data within their arms, avoiding the need to send information to the brain. In fact, the octopus has more than a majority of its neurons in its arms[2]. In machines, simple switches (e.g., a thermostat) can provide direct feedback control without a central processor, in which case the sensor is also the physical switch, yet a switch only provides a localized, yes/no response. Others have elegantly used materials-strategies, including origami[1], microfluidics[3,4], light-responsive compounds[5], and dielectric elastomers[6–8] to demonstrate logic, coupled dynamics, and feedback loops without the need for solid-state electronics. The approach here is distinguished by its simplicity, use of entirely soft materials, and ability to use sensory inputs (touch) to create visual outputs in which the logic occurs at the materials level without the need for semiconductors.

Here, we show that innervation of liquid metal (EGaIn, 75% Ga, 25% In by weight, melting point 15.7 °C)[9] within elastomers (polydimethylsiloxane, PDMS) can serve a variety of functions, including "soft tactile logic". Joule heating of the liquid metal can deliver thermal energy locally to thermochromic species within the elastomer for color changing passive displays, camouflage, and dynamic surfaces. These circuits benefit from the ability of the liquid metal to change geometry in response to strain and pressure[10,11]. This coupling of Joule heating to touch and deformation allows these dynamic composites to detect touch and strain. In more complex circuits, touch redistributes electrical current through these soft, embedded networks, inducing color change (a visual, digital output) via Joule heating. The color change can occur locally or distally depending on the design of the circuit. This concept is simple, avoids the use of a processor,

and utilizes completely soft materials with mechanical properties similar to tissue (and thus, should be compatible with stretchable electronics and soft robotics). Although the output here is color, this conceptual platform provides a mechanism for redistributing energy throughout a material.

## Results

**Establishing the platform**. We begin by demonstrating the basic platform as a tool to achieve color change. Natural organisms use color change for camouflage and expression, and do so using entirely soft materials[12,13]. Color change provides various benefits to different animal species such as the mandrill (for showing excitement), dart frog (to warn predators), and chameleons and cephalopods (for hiding from predators)[14–16]. Humans also rely on color change in high-tech electronics (monitors and displays) as a primary means of human–computer interfaces[17,18]. Color change can also communicate information, such as emotion in the case of skin tone (e.g., blushing with embarrassment or going pale with fear)[19,20].

Whereas displays (including stretchable light-emitting diodes (LEDs))[21] often use "active strategies" that rely on light generation, most animals utilize "passive strategies" in which ambient light (e.g., sun light) reflects off a surface to the human eye. A variety of strategies exist to replicate passive color change in synthetic systems, such as thermochromic liquid crystals[22,23] or pigments[24,25], colored fluids pumped through microchannels[26,27], thin film interference[28], dynamic photonic crystals and plasmonic structures[29–32], magnetically responsive materials[33], and electrochromic molecules[34]. Here, we use thermochromic pigments.

Figure 1a shows the basic platform. Liquid metal is patterned between two layers of PDMS, one transparent and one containing thermochromic species. The transparent layer is not critical for the device, but it allows visualization of the metal. Passing current through the liquid metal generates Joule heating and the thermochromic species alter their colors above critical temperatures due to the molecular structure rearrangement (molecular structures shown in Supplementary Fig. 1).

To demonstrate this principle, we mixed a temperature-sensitive rose red TF-R1 thermochromic species in PDMS (Sylgard-184 with 1.2 wt% red). This device shows red color initially, yet becomes white when the temperature exceeds 28 °C (see Supplementary Movie 1). An IR image (inset of Fig. 1b) shows the elevated-temperature regions, which corresponds to the observed visual pattern (Fig. 1b). The mechanism of color change is similar to that of leuco dyes[35–37]. The ability to create patterns of color suggests this approach may be useful for soft, passive displays.

This concept can work with a variety of colors. For example, we used blue thermochromics (0.8 wt% crystal violet lactone in Sylgard-186). This device is blue at room temperature but changes to white beyond 37 °C (see Fig. 1c and Supplementary Movie 2). Note that the device does not change color in the regions where we inserted copper wire connections because the resistivity of copper ($1.68 \times 10^{-6}\,\Omega\,cm$) is lower than the resistivity of liquid metal (EGaIn: $29.4 \times 10^{-6}\,\Omega\,cm$)[38], and therefore generates less Joule heating than EGaIn.

Since the thermochromic components have different response temperatures, mixing two species generates a new system that displays three color states, which offers a broader color pallet and more complexity (Fig. 1d). The temperature map in Fig. 1d indicates the ability to go from purple ($T < 28\,°C$), to blue (purple changes to blue in the absence of red, from $28\,°C < T < 37\,°C$), and finally white (in the absence of blue and red, $T > 37\,°C$). Figure 1d shows an example of such color change that occurs

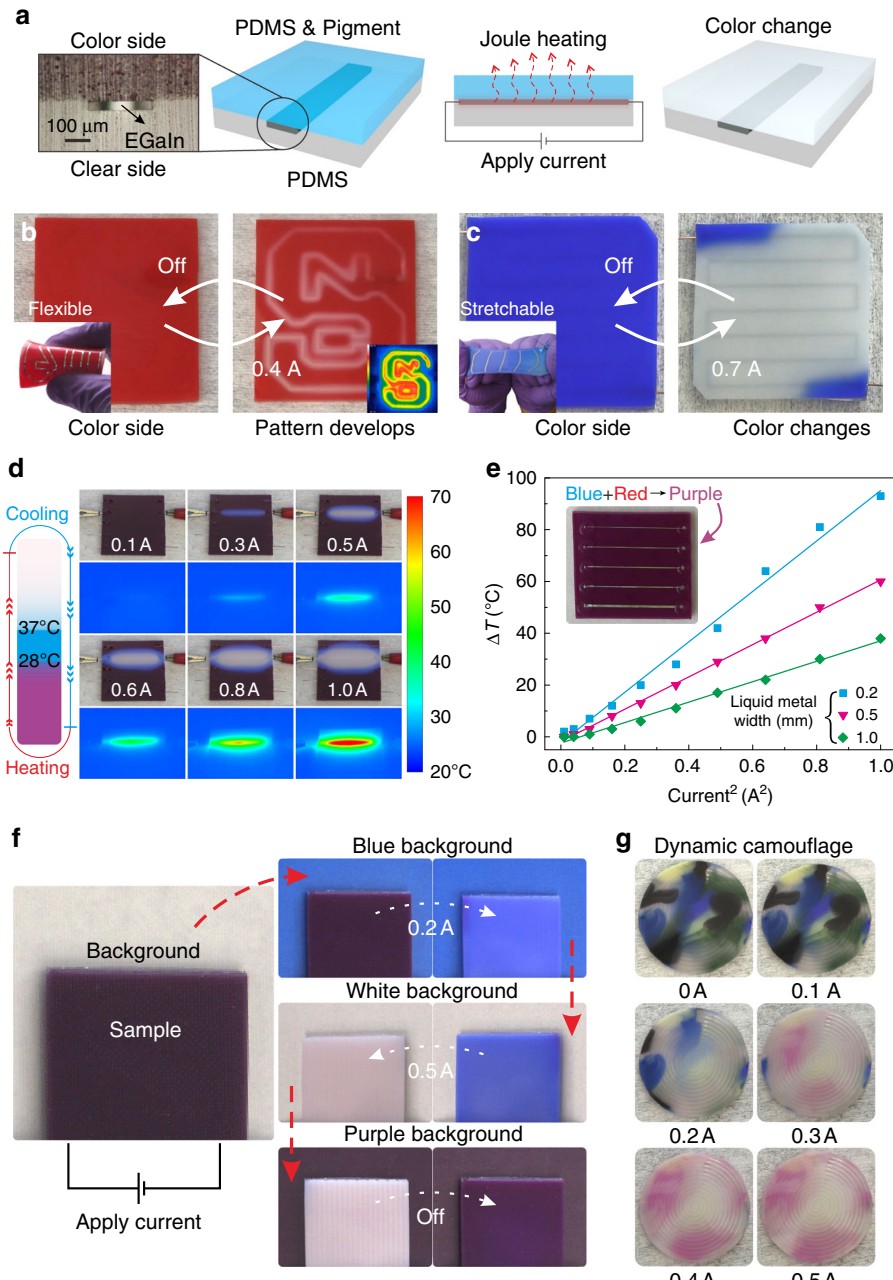

**Fig. 1** A platform for materials logic via pigmented silicones innervated with liquid metal. Liquid metals patterned within an elastomer can Joule heat to invoke color changes. **a** Schematic of a simple device consisting of a liquid metal circuit between two layers of PDMS. Joule heating causes color change of the pigment in the PDMS. **b** Localized Joule heating changes the color from red to white, resulting in a passive display. **c** A device with blue thermochromic completely changes color to white in response to electrical current. **d** Photographs of a sample with blue and red pigments that change color in response to current. The images under the photographs are IR images. **e** Plot of max temperature versus current squared ($\Delta T$ is surface temperature minus room temperature). The lines are best fits of the experimental data. The inset photograph shows the device with both red (1.2 wt%) and blue thermochromic (0.8 wt%). The dimensions of the linear microchannels are 50 mm length; 0.05 mm height; $X$ mm width ($X = 0.2, 0.3, 0.4, 0.5,$ and 1.0). **f** Modulating applied current allows the device to change colors to match the background. **g** Patterning dye in the elastomer can create multi-colored camouflage skins that change color in response to applied current

with increased current (see Supplementary Movie 3; the width of liquid metal wire is 0.4 mm).

To understand the power required to achieve these changes in color, we reasoned that Joule heating generates power ($P$) based on the applied current ($I$) and the resistance of liquid metal ($R$) according to $P = I^2R$. Since the resistance depends inversely on the width of the channel, we performed experiments in which we held all of the geometric parameters constant while varying the

width of the channels (Fig. 1e inset picture). We varied the current and measured the maximum surface temperature of the surface using an IR camera. As shown in Fig. 1e, the temperature is a linear function of the current squared for a given geometry, as expected (for more data, see Supplementary Figs. 2 and 3). For a given current, decreasing the width of the liquid metal increased the temperature change due to increased Joule heating (see Supplementary Figs. 4 and 5 for information on geometry).

Increasing the current not only increases the maximum surface temperature, but also increases the area experiencing elevated temperatures on the surface. We measured the width of the color change area of each device to construct the link between current and the width of color change area (see Supplementary Fig. 5). Increasing the current density—either through increased current or reduced width of the liquid metal—caused the color-changing regions to widen. The width of these regions agrees with thermal modeling done using COMSOL finite element simulations (see Supplementary Figs. 6 and 7; more information about modeling see Supplementary Note 1). Taken in sum, these results underscore the relation between geometry and Joule heating for a given current.

*Camouflage.* The ability to change color offers the possibility of camouflage. Many animals use camouflage to blend in with their surroundings[39,40]. Using the principles from Fig. 1a, we sought to demonstrate an elastomer that can change color to match the background. We put the device against various background colors (Fig. 1f). Adjusting the current through a serpentine of liquid metal adjusts the color to match the background (see Supplementary Movie 4). It is possible to use a broader pallet of colors by patterning the thermochromic species within the elastomer (Fig. 1g). While individual colors disappear sequentially with increased current, some thermochromic species reveal color with increased current (see Supplementary Movie 5).

**Thermomechanochromism.** The use of liquid metal for Joule heating is interesting because the dimensions of the conductor change during deformation. The ability of liquid metal wires to change shape, and thus resistance during elongation offers the opportunity to create dynamic Joule heating properties. Thus, these circuits can thermochromically report on the state of deformation. This approach has similarities to mechanochemistry, in which molecules are designed to react (and often change color) in response to stress. Mechanochemistry has been utilized in soft robotics and soft electronics to achieve color change and warn users of impending mechanical failure[41,42]. The system here can do the same, but does not require any special chemistry and has multiple colorimetric outputs, whereas mechanochemistry typically only provides a single color change at a single stress (or strain). Because color change here is driven thermally, yet is triggered mechanically, we refer to it as thermomechanochromism.

As a simple implementation, straining a liquid metal wire causes its length to increase, while simultaneously decreasing its cross section area. When the cross-section of a wire narrows, the resistance increases and thus the Joule heating increases, as shown conceptually in Fig. 2a. According to geometric considerations, strain ($\varepsilon = (L - L_0) \, L_0^{-1}$, where $L$ is the length) enlarges the resistance from its initial value ($R_0$) following Eq. (1):

$$R = R_0(\varepsilon + 1)^2. \tag{1}$$

We utilized this special feature to create a sensor that changes color in response to strain. The resistance of a liquid metal wire increases with elongation, as expected (Fig. 2b). We applied a constant current of 0.2A while stretching (see Supplementary Movie 6). This current is insufficient to cause color changes at zero strain, but as the wire elongates, the Joule heating increases and color change occurs. Figure 2c shows photographs of 0.2 mm wide liquid metal wires at various strains. The device changes from purple to blue due to the absence of red coloration (at temperatures >28 °C) and it further switches from blue to white at 60% strain due to the activation of the blue thermochromic components. The device becomes purple again when returned to

0% strain due to the reversible thermochromism. This demonstration shows the possibility of achieving multiple, step-wise color changes via stretching and also sensing strain via the width (and color) of the color profile. Whereas others have realized strain-sensitive color changing materials using molecular strategies to achieve color change (e.g., mechanochemistry or block copolymers)[41,43], here the strategy has the appeal of being more general since it does not rely on mechanochemistry. It does, however, require energy dissipation (electrical current).

We sought to understand how the color changes with strain and demonstrate the ability to tune the color response to strain. Inserting Eq. (1) into $P = I^2 R$ and taking the derivative gives Eq. (2):

$$\frac{dP}{d(\varepsilon + 1)} = 2I^2 R_0(\varepsilon + 1). \tag{2}$$

Eq. (2) shows that the change in power ($P$) with strain ($\varepsilon$) depends on the current ($I$, which is constant in these experiments) and the initial resistance. Thus, Eq. (2) predicts that liquid metal wires with higher initial resistance will be more responsive to stretching. To confirm this expectation, we fabricated linear conductors 35 mm long, 0.05 mm tall and $X$ mm wide ($X = 0.2, 0.3, 0.4, 0.5,$ and $1.0$) and then measured the width of the color change area while applying constant current (see Supplementary Fig. 8a). The trends in Supplementary Fig. 8a confirm that devices with narrower liquid metal wires change change color at lower strains. Conversely, the widest device never becomes white even at 180% strain because of its low initial resistance. This result indicates that it is possible to tune the device by adjusting initial resistance so that the color change happens at a particular value of strain (here, we adjust initial resistance by changing the width of the channel, but the height could also be varied). In contrast, molecular strategies typically evoke a single color change a specific stress or strain (defined by the mechanochromic molecules themselves, such as spiropyran).

Current is another factor which affects the color change responsiveness, according to Eq. (2). We applied currents of 0.2, 0.3, and 0.4 A on a device with 1 mm wide liquid metal (see Supplementary Fig. 8b). The device does not change to white during elongation using 0.2 A. After increasing the current to 0.4 A, the device can change color at low strain (≈50%). This observation confirms that it is also possible to use current to control the strain where color change occurs.

**Soft tactile sensing.** Compression is another way to change the dimensions of microfluidic channels. This type of deformation has been used previously to create touch sensors that electrically sense changes in the resistance or capacitance of liquid metal wires[10,11,44,45]. We reasoned the changes in the cross area of the microfluidic channel arising from compression could induce local color change by changing the local resistance (if the current is held constant). Geometric considerations predict resistance should scale with cross sectional area as Area$^{-1}$. We applied 0.1 A current to a device and pressed it with 100, 200, 300, and 400 kPa pressures at intervals of 15 s over an area of $1 \times 1$ cm$^2$. The device changed color locally where we applied pressure. The pressed area changed from purple to blue at 100 kPa. The compressed area changed to white when the pressure was 200 kPa (see Supplementary Movie 7). In principle, tuning the modulus of the elastomer could change the pressure sensitivity.

We sought to demonstrate how the current and the width of liquid metal affects the color response to compression. We fabricated a microfluidic channel 50 mm long, 0.05 mm tall, and $X$ mm wide ($X = 0.2, 0.3, 0.4, 0.5,$ and $1.0$). We applied 0.1, 0.2, and 0.3 A current on a device with a 1 mm wide liquid metal trace

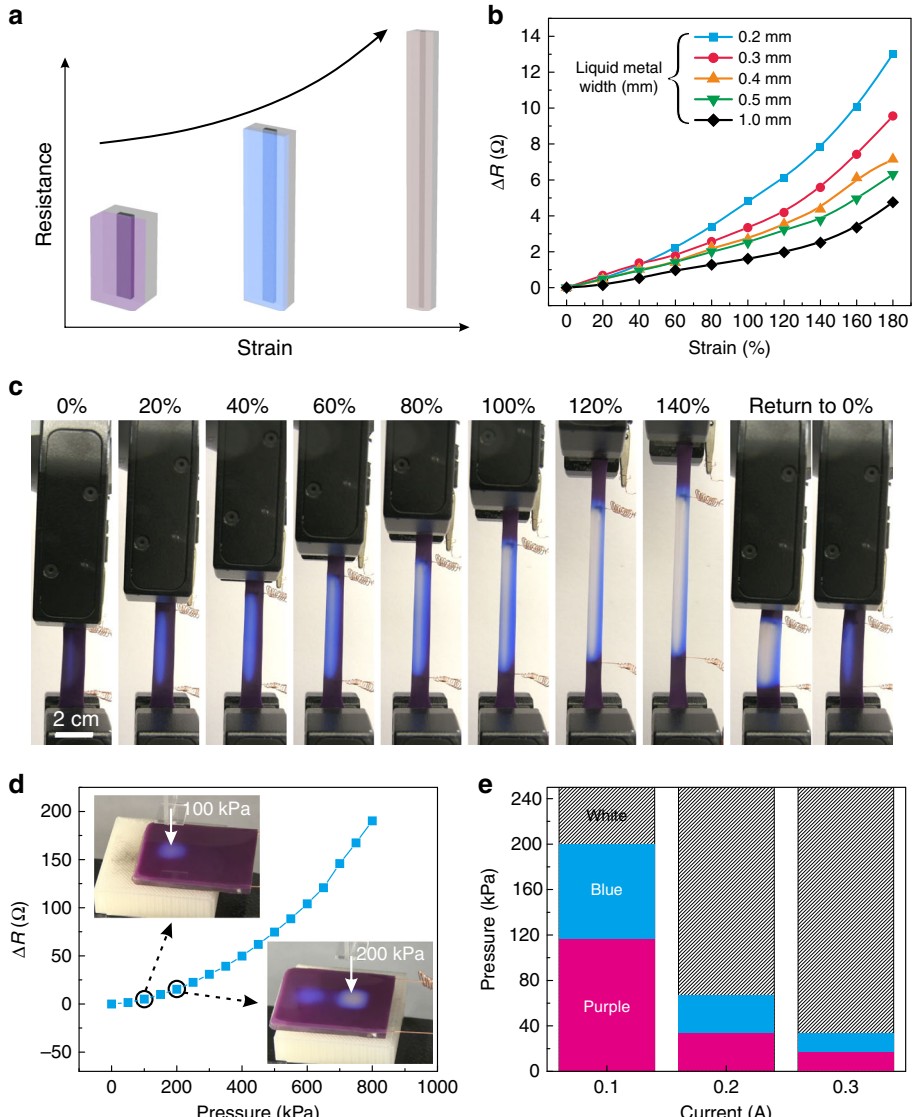

**Fig. 2** Thermomechanochromism effects and tactile sensing. **a** Stretching a liquid metal wire causes the geometry of the wires to change, and thus, the resistance and the current density increase. These changes increase Joule heating, resulting in a color change. **b** Change in resistance as a function of strain. $\Delta R$ is the resistance at a given strain minus the resistance at 0% strain. **c** Photographs of a device being stretched while maintaining constant current (but increased current density). **d** Change in resistance as a function of pressure through a single serpentine conductor. Inset images show the color change in response to different pressures (100 and 200 kPa). **e** The color response depends on the current and pressure for a 1 mm width linear microchannel device

and measured the color as a function of pressure (Fig. 2e). The ranges of pressure values where purple and blue exist narrow with increased current.

**Soft tactile logic using materials-based principles**. In Fig. 2d, e, there is a single conductive path. Thus, pushing the circuit changes the local resistance and increases the local current density since the current must pass through the depressed region. However, it is also possible to design circuits with multiple paths for current. This concept can be utilized to redistribute energy in a circuit and achieve simple logic operations without the use of semiconductors.

We fabricated a circuit to demonstrate "soft tactile logic" (Fig. 3a). This device consists of two "input" areas (labeled as A and B in Fig. 3a) and one "display" area (labeled as C). Here, the input signals are pressure. Similar to a threshold voltage in a transistor, there is a threshold pressure required to induce a

sufficient current to cause color change in the "display region". Following the language of binary logic, pressures above a threshold are called "1" and pressures below are a "0". We applied 0.4 A current and input various "1" and "0" pressure signals to regions A and B. Region C responded to the input by changing the color between purple, blue, or white (see Fig. 3b and Supplementary Movie 9). These changes in color are due to the redirection of electrical current to Region C based on the physical input (see Supplementary Fig. 10). This simple device is a NAND-like logic operation, although the output is complicated by three output states rather than two. It is possible to extend this concept to parallel 'shunt' circuits that redistribute current when pressed (see Supplementary Fig. 9, Supplementary Movie 8 and Supplementary Note 2). In addition to causing Joule heating, the redistributed current can also turn on circuit elements (such as LEDs, as shown in Supplementary Movie 10 and Supplementary Fig. 11) or mechanical elements (such as a fan, as shown in Movie 11 and Fig. 12).

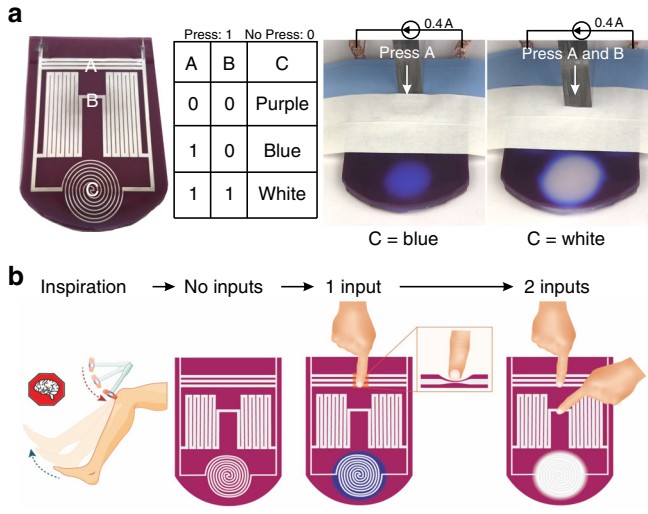

**Fig. 3** Soft tactile sensors and logic via energy redistribution. **a** The underside of a soft tactile logic device. Pressing (1) or not pressing (0) on regions A and B can redirect current to region C to evoke a distal change in color in a manner that follows a truth table. **b** Mechanism of soft tactile logic. A reflexive, "knee-jerk" reaction bypasses the human brain. Likewise, the distal color change does not require a centralized processor. Instead, the response arises due to redistribution of the electrical energy using materials-based logic

## Discussion

The results of Fig. 3 are interesting because the logic and response are built entirely within these soft materials at the local level without the need for conventional semiconductor-based logic elements. We acknowledge that this is a modest concept demonstrator that is limited in its present form. Nevertheless, it is thought provoking to consider the concept further.

Although the concept has similarities to a variety of conventional devices and systems, it draws no perfect comparison. It is similar to a variable resistor (as well as a potentiometer, transducer, current divider, and even some original mechanical computing machines), yet it is distinguished by being built entirely from soft materials, multiplexed to achieve logic-like output, and converts analog information (pressure and current) into a "digital" visual output (which is inherently useful since humans readily interact with systems through visual cues). For these reasons, we chose the term "soft tactile logic". It is, however, limited as a logic device in its present form because the input is mechanical, yet the output is electrical and optical (and thus, the output cannot be readily input into another circuit element).

Whereas the current here generates color change (either locally or distally, depending on the design)—which may find use in interactive displays or camouflage—the output (current) could be utilized for other purposes, such as for powering electronics (cf. Movie 10), actuators (cf. Movie 11), or stiffening mechanisms. It could also be used for local control loops to reroute electrical current in response to mechanical deformation. Thus, the ability to perform local, materials-based "morphological computation" may be useful to better control the complex body dynamics that occur in soft actuators (i.e., nonlinearity and high degrees of freedom). Similar to an octopus, there is both centralized information (in the form of applied current to the device), as well as localized information. The system also has some similarities with the reflex response loops used by humans and animals to bypass the need for central processing. The concept also has some similarities to a Venus fly trap, which utilizes "integral" inputs to determine whether it should snap shut (in other words, a single touch does not initiate the snapping, but rather, multiple

touches)[46,47]. This type of response, called "fading memory", is defined as "the property of a system that retains the influence of a recent input sequence within the system, which permits the integration of stimulus information over time"[48]; here, the integral information is heat. Finally, it has some similarities to "embodied logic", in which the "body" (here, the elastomer and metal) itself gives the material its "intelligence"[2].

Finally, the use of thermochromics does have several limitations. First, the color is triggered by temperature, thus its behavior will change depending on the baseline temperature (here, all measurements started at room temperature). Other applications of soft tactile logic will not have this issue. Second, the current necessary to cause Joule heating is significant, although we made no effort to optimize the Joule heating. Finally, due to thermal transport, the response is quite slow (seconds to tens of seconds) and the resolution of the 'images' are diffuse. This could be optimized or manipulated by varying the thermal transport properties of the encasing materials. There are some simple ways to increase the response time. Using silicones with high thermal conductivity can increase heat transfer[49]. Optimizing the geometry of thermochromic elastomers is another option to increase the response time. For example, thinner samples change temperature faster (see Supplementary Fig. 12 and Supplementary Movie 12).

In conclusion, this paper describes soft and stretchable color changing composites that serve as soft camouflage, thermomechanochromic strain sensing and soft tactile logic. The platform is simple, composed of only liquid metal, PDMS, and thermochromic species. In its simplest implementation, passing an electrical current through liquid metal generates Joule heating and consequently changes the color of the thermochromic components. Combining multiple thermochromic species enables a broader range of color possibilities including the ability to make dynamic coloration for passive displays or camouflage. Stretching or pressing these devices changes the cross-sectional geometry of liquid metal and therefore changes the resistance. The device can, therefore, serve as a stress (or strain) sensor by transforming the stress change to color change through variation of electrical resistance. More complex circuits can redirect electrical energy in response to touch to create soft tactile logic. The touch input (analog) generates color (digital), which can occur locally or distally via a threshold response to the input pressure. Because of the soft and stretchable properties of composites, they could potentially serve as e-skins to work with soft robots, prosthetics, and stretchable electronics.

The concept of soft tactile logic is particularly interesting because it avoids the conventional approach of using sensor arrays that connect to a central processor to achieve a response. Instead, the logic is built into the materials themselves without the need for external semiconductor devices. This offers the possibility of "distributed logic". In addition, the response can be local or distal. Here, we demonstrated color change as the output, but the ability to redirect current through soft "nerve networks" could be used for other purposes, such as actuation, stiffening, adhesion, or feedback control. In addition, we show only the response to touch and strain but other modes of deformation may invoke unique coupled responses, such as twisting, swelling, or inflating the elastomer.

## Methods

The PDMS microfluidic channels were fabricated using soft photolithography from a commercial PDMS elastomer kit (Sylgard-184 and Sylgard-186, Dow Corning). The transparent PDMS microfluidic channels were obtained by molding against a topographically patterned silicon wafer (average PDMS thickness: 1.80 mm). The colored layers of PDMS containing thermochromic species were obtained by peeling silicone from a flat dish (average thickness: 1.21 mm). After exposure to oxygen plasma, we used the sheets of thermochromic species-loaded PDMS and

plain PDMS to produce a microfluidic channel into which we vacuum-filled the liquid metal[50]. Uncured PDMS sealed the inlet and outlet holes of the channel to assist with handling. Commercial thermochromic pigments were purchased from Shenzhen Qiansebian Pigments Co., Ltd. The thermochromic mechanism was described in the Supplementary Note 3. These pigments maintained their color "fastness" through at least 500 thermal cycles (Supplementary Fig. 14). We used an extensometer (Instron 5943) to perform strain and compression tests while measuring force. We used an IR camera (FLIR SC300-series) to record the surface temperatures. All the photographs and Movies were recorded by a camcorder (Canon VIXIA HF G20). The patterns of all the circuits are shown in Supplementary Fig. 15.

## Data availability

The authors declare that the data supporting the findings of this study are available within the paper [and its supplementary information files]. Data is available upon request.

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

## Acknowledgements

The authors gratefully acknowledge helpful conversations with colleagues, including Alper Bozturk, Jan Genzer, Alon Gorodetsky, Denys Makarov, Rob Shepherd, Sam Stanton, and Rich Vaia. The authors thank the CSC program for support and to the NSF (CBET-1510772) and ERC ASSIST center (ERC EEC-1160483).

## Author contributions

Y.J., Y.L. and M.D. designed the project. Y.J. performed the material synthesis, device fabrication, and testing. Y.L. fabricated the devices. A.K. performed the COMSOL simulations. I.J. helped the record and edit videos as well as perform strain tests. M.G. supplied materials and advised on appropriate pigments. Y.J., Y.L. and M.D. co-wrote the paper. All authors (Y.J., Y.L., A.K., I.J., M.G. and M.D.) discussed the results and commented on the paper.

## Additional information

**Competing interests:** The authors declare no competing interests.

