## [Peer Review File · Nature Communications]

Reviewers' comments:

Reviewer #1 (Remarks to the Author):

This work presents a passive display based on the phenomena of color change in thermochromic elastomers by leveraging Joule heating of embedded liquid metal wires.

The design and manufacturing of the device are presented and its functionality is demonstrated to different modes of operation: Mechanical deformation(mechano-chromic), localized pressure (tactile sensing) and soft tactile logic.

Overall the paper is easy to read and motivates the reader into the development of distributed computation at the soft material level that does not rely on semiconductors.

I believe it presents two main contributions: i. The use of liquid metal Joule heating for color change in thermochromic elastomer that responds to mechanical inputs and can be controlled by current. ii. the introduction and demonstration of the concept of soft tactile logic.

I encourage the authors to deepen the exploration of soft tactile logic due to its relevance in the soft robotics community. In addition, I recommend to reframe the part where the authors state that their system breaks the 'sense-compute-respond' paradigm because this concept is still applicable in the current implementation -computation is defined as any processing of information and nevertheless the current system does not rely in traditional computation, one can argue that the computation is done at the material level.

Reviewer #2 (Remarks to the Author):

This is rather an interesting study on using liquid metal elastomer composites with thermochromics response. The color change in these composites is activated through Joule heating and it is claimed the collected signal for these composites can be used for more useful functional applications such as sensing of touch and pressure in a completely soft electronic system. The importance of this study is based on demonstrating the concept of "sense-compute-respond" in synthetic smart devices. The paper is well organized with clear figures and supporting videos. It is well written and the discussion part is comprehensive. The amount of new data presented in this manuscript is sufficient but I'm not sure about the novelty of this work and possible broad impact of the study in order to be considered for publishing in Nature Communications journal. There are a few other comments that should be addressed by the authors in order to improve the manuscript and make it publishable in a journal article.

Major Comments:

1. It seems the role of liquid metal is particularly improving the heat transfer in the elastomer (obviously because of the high thermal conductivity of the liquid metal) and the output is just a color change. What part is really new? And what is the main and impactful contribution of this study?

2. A lot of data is presented on changing the resistance of the channels by reducing the channel width and therefore increasing the resistance will increase the temperature because of the Joule heating effect. Figure 1e, as well as Figure S2-S3-S4-S5-S6 which includes the COMSOL finite element simulations. This simple calculation seems a try to add more data to the manuscript without providing any new insight or useful conclusions.

3. The "Mechanochromics" effect showing in the paper sound interesting. However, it is exactly

what was expected. Changing the resistance, then changing the amount of heat generated by Joule heating. Compare to this material, we have Mechanoluminescence effect that can be generated without the use of Joule Heating. What is the main benefit of using liquid metals here? And what is new beside the Joule heating?

4. The response time of the fabricated devices is very low. It takes a while to cool down and see observable color change. Is there any way to increase the response time?

5. According to the title of this manuscript "Distributed Tactile Logic via Innervated Soft Thermo-chromic Elastomers" there should be distributed tactile sensors. However the final device has only 2 inputs at the end (Figure 3). The word "distributed" is misleading in the title. The authors may consider revising the title.

Reviewer #3 (Remarks to the Author):

This is an interesting article on thermochromism. There is a lot of recent developments that are ignored and non mentioned by the authors. The proposed technique to change color is quite innovative and has to be considered. However, the considered thermo-chromic materials exhibit a very fast ageing and loose their properties quite fast. Thus, the use of the whole technique without solving the ageing problem is of a limited importance.

Once the issue is considered in a credible way, the paper deserves to be considered for publication

Reviewer 1

This work presents a passive display based on the phenomena of color change in thermochromic elastomers by leveraging Joule heating of embedded liquid metal wires.

The design and manufacturing of the device are presented and its functionality is demonstrated to different modes of operation: Mechanical deformation(mechano-chromic), localized pressure (tactile sensing) and soft tactile logic.

Overall the paper is easy to read and motivates the reader into the development of distributed computation at the soft material level that does not rely on semiconductors.

I believe it presents two main contributions: i. The use of liquid metal Joule heating for color change in thermochromic elastomer that responds to mechanical inputs and can be controlled by current. ii. the introduction and demonstration of the concept of soft tactile logic.

I encourage the authors to deepen the exploration of soft tactile logic due to its relevance in the soft robotics community.

Response:

Thank you very much for your time and positive feedback. We spent significant time writing the manuscript to make sure we balanced reporting on the various novel aspects, which you have identified succinctly. We believe the soft tactile logic may be the most impactful aspect of the work. Our intention was to introduce the concept of ‘soft tactile logic’ as a portion of the paper to serve as inspiration for those working on soft robotics (to be frank, we are just on the periphery of this field). To this end, we have included some vision of the possibilities (and limitations) in the outlook portion of the paper.

Per your request, we have deepened the exploration of the concept of ‘soft tactile logic’ by including two more demonstrations. In one demonstration, a LED turns on only after two soft conductors are pressed (Figure S10, Video 10). In another demonstration, a fan turns on when touched once, but off if touched in two places (Figure S11, Video 11). We hope that these simple demonstrations help extend the concept of ‘soft tactile logic’ beyond color change and into electronics and mechanics.

Supplementary Figure S10

In addition, I recommend to reframe the part where the authors state that their system breaks the ‘sense-compute-respond’ paradigm because this concept is still applicable in the current implementation - computation is defined as any processing of information and nevertheless the current system does not rely in traditional computation, one can argue that the computation is done at the material level.

Response:

Thank you for the excellent suggestion.

Previously, we used that phrase twice in the manuscript (abstract and discussion). We have replaced it with text describing ‘computation at the materials level’ in the absence of semiconductor-based logic. We have also tweaked the introduction to capture the same concept.

Reviewer 2

This is rather an interesting study on using liquid metal elastomer composites with thermochromics response. The color change in these composites is activated through Joule heating and it is claimed the collected signal for these composites can be used for more useful functional applications such as sensing of touch and pressure in a completely soft electronic system. The importance of this study is based on demonstrating the concept of “sense-compute-respond” in synthetic smart devices. The paper is well organized with clear figures and supporting videos. It is well written and the discussion part is comprehensive. The amount of new data presented in this manuscript is sufficient but I’m not sure about the novelty of this work and possible broad impact of the study in order to be considered for publishing in Nature Communications journal. There are a few other comments that should be addressed by the authors in order to improve the manuscript and make it publishable in a journal article.

Response:

Thank you very much for your time and thoughtful comments. We appreciate the kind words about the work. We are excited about it and have informally gotten the same response from a small group of scientists with whom we have privately shared this work.

We believe the two most novel aspects of the work are (1) the coupling of physical deformation / touch / strain to distribution of electrical energy (reported here primarily via color change, but could extend to electronics and mechanics, as we now show through new demonstrations), and (2) the demonstration of logic at the materials level that can respond to touch.

Regarding the former, the most relevant literature is found in the field of mechanochemistry, in which color changes at a strain value that cannot be easily tuned (color change occurs at strains defined by the molecules themselves). The approach here can tune the strain at which the color response occurs using geometric principles, rather than chemistry.

Regarding the latter, there are several examples of unconventional logic, but none have the combination of being built from soft materials and the ability to do ‘computation’ at the local level.

We hope that taken in sum, these contributions, combined with the ability to create passive ‘displays’ will be of interest to the broad community reached by this interdisciplinary journal.

1. It seems the role of liquid metal is particularly improving the heat transfer in the elastomer (obviously because of the high thermal conductivity of the liquid metal) and the output is just a color change. What part is really new? And what is the main and impactful contribution of this study?

Response:

We agree that the liquid metal has a much higher thermal conductivity relative to the silicone. We note, however, that the metal itself is a miniscule fraction of the overall device volume, so it likely has a minimal role on the thermal conductivity. Instead, the liquid metal is utilized for Joule heating. The Joule heating causes color change. There are many ways to invoke color change in materials, so color change by itself is indeed not new. The primary novelty relates to the coupling of mechanical deformation to the distribution of electrical energy for sensing strain and performing simple logic in

response to touch at the materials level. We use color as a way to ‘report’ the response because of the importance of displays for human-device interactions. The first portion of the manuscript focuses on establishing the basic platform, whereas the latter part of the manuscript relates to the role of mechanical deformation via strain or touch.

2. A lot of data is presented on changing the resistance of the channels by reducing the channel width and therefore increasing the resistance will increase the temperature because of the Joule heating effect. Figure 1e, as well as Figure S2-S3-S4-S5-S6 which includes the COMSOL finite element simulations. This simple calculation seems a try to add more data to the manuscript without providing any new insight or useful conclusions.

Response:

Thank you for the comment. We would be happy to remove data if there is some you find to be unnecessary. As the reviewer kindly noted, most of the data is in the supporting information. We include it to add rigorous characterization due to the importance of geometry on the Joule heating and the role of mechanical coupling to the color output. We could delete Figure S3 and perhaps Figure S2, and merge Figures S5 and S6, but since these are all in the supporting information, we have tentatively left them “as is”.

3. The “Mechanochromics” effect showing in the paper sound interesting. However, it is exactly what was expected. Changing the resistance, then changing the amount of heat generated by Joule heating. Compare to this material, we have Mechanoluminescence effect that can be generated without the use of Joule Heating. What is the main benefit of using liquid metals here? And what is new beside the Joule heating?

Response:

We agree that this concept is simple and embrace the simplicity in its elegance. We have some experience with mechanochemistry (Barbee, M. H. et al. Mechanochromic Stretchable Electronics. ACS Applied Materials & Interfaces 10, 29918–29924 (2018)). In our humble opinion, there are two limitations: first, it requires chemistry – both the facilities and experience to carry out the chemistry, but also the correct chemical functional groups to bind the chromophore into the polymer network. In contrast, the approach here requires no chemistry and can be incorporated into a wide range of materials. Second, the most common functional group (spiropyran) changes a single color at a single strain and with a single color. The approach here can tune the color and the strain at which color change occurs. We have noted the downside in the manuscript: the approach here requires energy input. In that regard, mechanochemistry is also elegant in its own right.

4. The response time of the fabricated devices is very low. It takes a while to cool down and see observable color change. Is there any way to increase the response time?

Response:

This is an excellent observation. The silicones here have not been optimized for heat transfer. By using silicones with high thermal conductivity, it should be possible to transfer heat faster. For example, adding boron nitride (Zhou, W.-Y., et al. “Thermally conductive silicone rubber reinforced with boron nitride particle” *Polymer Composites* 28, 23–28 (2007)) can improve the thermal conductivity of silicone by ~ 7x and recent work shows liquid metal can make the thermal conductivity

approach that of stainless steel (Bartlett, M. D. et al. “High thermal conductivity in soft elastomers with elongated liquid metal inclusions”. *PNAS* 114, 2143–2148 (2017).)

Another option is optimizing the geometry. We have added a supporting video to show how thickness changes the response time (see Supplementary Figure S12 and Supplementary video 12). Active heating / cooling is also possible, such as using heat transfer fluids or thermoelectrics, but this raises the complexity of the system. In the manuscript, we have noted the importance of optimizing thermal conductivity for improved performance as future work.

Supplementary Figure S11 The cooling time of thermochromic elastomers with different thickness.

5. According to the title of this manuscript “Distributed Tactile Logic via Innervated Soft Thermochromic Elastomers” there should be distributed tactile sensors. However the final device has only 2 inputs at the end (Figure 3). The word “distributed” is misleading in the title. The authors may consider revising the title.

Response:

Thank you for the suggestion. To be forthright, prior to submission we discussed many different titles that would be concise yet capture the essence. We are open to suggestions. We used “distributed” in reference to the fact there is not a centralized processor, but rather, the logic can be “distributed”.

We have tentatively changed it to “Materials Tactile Logic via Innervated Soft Thermochromic Elastomers”, but remain open to suggestions.

Reviewer 3

This is an interesting article on thermochromism. There is a lot of recent developments that are ignored and non mentioned by the authors. The proposed technique to change color is quite innovative and has to be considered. However, the considered thermochromic materials exhibit a very fast ageing and loose their properties quite fast. Thus, the use of the whole technique without solving the ageing problem is of a limited importance.

Once the issue is considered in a credible way, the paper deserves to be considered for publication

1. There is a lot of recent developments that are ignored and non mentioned by the authors.

Response:

Thank you very much for your time and insights.

We did not mean to slight any work from the literature. We prioritized the discussion in the paper to make it clear that the focus is less about thermochromism, and more about coupling deformation (strain, touch, etc) to energy distribution in a soft material. Nevertheless, we have cited seventeen references on color change (including a 2018 review - ref 14, those below, 29-34, and out of respect for the reviewer, one from 2019: ref 27). We would be happy to include more if there are specific ones suggested, although we are approaching the limit for references provided by the journal:

“A variety of strategies exist to replicate ‘passive’ color change in synthetic systems such as thermochromic liquid crystals^{16,17} or pigments,^{18,19} colored fluids pumped through microchannels,^{20,21} thin film interference²², dynamic photonic crystals and plasmonic structures^{23–26}, magnetically responsive materials²⁷, and electrochromic molecules.²⁸”

2. However, the considered thermochromic materials exhibit a very fast ageing and loose their properties quite fast. Thus, the use of the whole technique without solving the ageing problem is of a limited importance.

Response:

Thank you for the helpful comment. We performed several additional experiments to show that the color change is repeatable and does not age over 500 cycles. We added Figure S12 to the supporting information to show the color “fastness” after 500 cycles of heating and cooling.

Supplementary Figure S13. Fastness test of a thermochromic elastomer after 500 cycles.

Reviewers' comments:

Reviewer #1 (Remarks to the Author):

In this paper, authors present innervated thermochromic elastomers with the capability of soft tactile distributed sensing and logic. In particular, the work demonstrates the color changing capability of elastomer using thermochromic pigments using joule heating principle and soft tactile logic using variable resistance. The presented idea is interesting and novel, and does deserve publication in this journal after addressing some questions.

My comments:

1) The article shows an interesting principle of what they call mechanochromism but I think the term is not totally correct, although the system behaves in a similar manner of it. The color change is still happening due to Joule heating effect not due to stress. Reviewer would suggest calling it dynamic thermochromics instead of mechanochromics. Or thermo-mechanic-chromism. Whatever the authors think, but at least representing the intermediate step required.

2) How did authors derive Eq. 1, if not please add a citation. Also, after taking the derivative of Eq. 2 please remove the constant from the denominator. 3) It would be interesting to plot the fig 1 e correspond to resistance instead of metal width because resistance is the key parameter. Reviewer suggests adding a characterization graph between resistance vs. metal width at different currents in the elastomer. 4) When authors talked about tactile sensing, but they did not discuss sensor's characteristics such as repeatability, accuracy, and hysteresis, etc. 5) Please add a citation for line 86-8, 90-92 and correct spacing in line 207.

Reviewer #2 (Remarks to the Author):

The authors have successfully addressed all the comments.

Reviewer #3 (Remarks to the Author):

Thanks a lot for the revised version. The issue of ageing is not adequately addressed. Information on the absorbed radiation, UV, etc), has to be given. The description provided is very poor. I am not interested to include any specific reference, but the authors do not have a clear knowledge of the recent developments on thermochromism.

Reviewer 1

In this paper, authors present innervated thermochromic elastomers with the capability of soft tactiledistributed sensing and logic. In particular, the work demonstrates the color changing capability of elastomer using thermochromic pigments using joule heating principle and soft tactile logic using variable resistance. The presented idea is interesting and novel, and does deserve publication in this journal after addressing some questions.

1) The article shows an interesting principle of what they call mechanochromism but I think the term is not totally correct, although the system behaves in a similar manner of it. The color change is still happening due to Joule heating effect not due to stress. Reviewer would suggest calling it dynamic thermochromics instead of mechanochromics. Or thermo-mechanic-chromism. Whatever the authors think, but at least representing the intermediate step required.

Response:

Thank you for the suggestion. We agree that both heat and mechanics are important.

Previously, we used that word four times in the manuscript. We have replaced it with 'thermo-mechanochromism'.

2) How did authors derive Eq. 1, if not please add a citation. Also, after taking the derivative of Eq.2 please remove the constant from the denominator.

Response:

Thank you for your suggestion. We have added a citation and also provide the derivation here for convenience. We left equation 2 "as is" because the math is valid if $(\epsilon + 1)$ is replaced by "y".

$$\epsilon = \frac{(L-L_0)}{L_0} \quad (1)$$

Convert Ep. 1 get Ep. 2

$$L = L_0 (\epsilon + 1) \quad (2)$$

$$R = \rho \frac{L}{A} \quad (3)$$

$$A = \frac{V}{L} \quad (4)$$

Merge Ep.3 and Ep.4 to get Ep.5

$$R = \rho \frac{L^2}{V} \quad (5)$$

Merge Ep.2 and Ep.5 to get Ep.6

$$R = \rho \frac{L_0^2(\epsilon+1)^2}{V} \quad (6)$$

$$R_0 = \rho \frac{L_0^2}{V} \quad (7)$$

Merge Ep.6 and Ep.7 to get Ep.8

$$R = R_0 (\epsilon + 1)^2 \quad (8)$$

3) It would be interesting to plot the fig 1 e correspond to resistance instead of metal width because resistance is the key parameter. Reviewer suggests adding a characterization graph between resistance vs. metal width at different currents in the elastomer.

Response:

Thank you for the suggestion. We plotted it versus width to show how the response can easily be tuned using geometry. To address the suggestion, we have added the following figure (S2) to relate the width to the resistance, which makes the analysis more general.

4) When authors talked about tactile sensing, but they did not discuss sensor's characteristics such as repeatability, accuracy, and hysteresis, etc.

Response:

Thank you for the comment. To address this, we have cited ref 10-11, which provide more rigorous characterization of liquid metal sensors. Similar to this previous work, we also found the response to touch to be very reproducible due to the elastic nature of the silicone.

5) Please add a citation for line 86-8, 90-92 and correct spacing in line 207.

Thanks for your comments. We have addressed it. In addition, we added two relevant citations about soft materials and camouflage in line 86-87.

Xu, C., Stiubianu, G. T. & Gorodetsky, A. A. Adaptive infrared-reflecting systems inspired by cephalopods. *Science* **359**, 1495–1500 (2018).

Stuart-Fox, D. & Moussalli, A. Camouflage, communication and thermoregulation: lessons from colour changing organisms. *Philos. Trans. R. Soc. Lond. B Biol. Sci.* **364**, 463–470 (2009).

We also added two relevant citations about color and human-computer interfaces in line 90.

Ye Zhang & Jia Wang. Human-computer interaction design strategy for color identification on multi-media visual interface. in *2009 IEEE 10th International Conference on Computer-Aided Industrial Design Conceptual Design* 1874–1877 (2009). doi:10.1109/CAIDCD.2009.5375081

Thurlings, M. E., van Erp, J. B. F., Brouwer, A.-M., Blankertz, B. & Werkhoven, P. Control-display mapping in brain-computer interfaces. *Ergonomics* **55**, 564–580 (2012).

And two citations about emotions, tone, and facial color in line 92.

Nakajima, K., Minami, T. & Nakauchi, S. Effects of facial color on the subliminal processing of fearful faces. *Neuroscience* **310**, 472–485 (2015).

Buechner, V. L., Maier, M. A., Lichtenfeld, S. & Elliot, A. J. Emotion Expression and Color: Their Joint Influence on Perceived Attractiveness and Social Position. *Curr. Psychol.* **34**, 422–433 (2015).

Reviewer 2

The authors have successfully addressed all the comments.

Reviewer 3

Thanks a lot for the revised version. The issue of ageing is not adequately addressed. Information on the absorbed radiation, UV, etc), has to be given. The description provided is very poor

Response:

Thank you for the helpful comment.

We performed several additional experiments to show that the color change is repeatable and does not age over 500 cycles. We added Figure S14 to the supporting information to show the color “fastness” after 500 cycles of heating and cooling. **To quantify the ageing of the pigment color, we report the absorptivity of the red and blue elastomers at different temperatures. b The absorptivity of red elastomers at 25°C (solid line) and 30°C (dotted line). c The absorptivity of blue elastomers at 25 °C(solid line) and 40 °C(dotted line).**

Supplementary Figure S14. Fastness test of a thermochromic elastomer after 500 thermal cycles.

I am not interested to include any specific reference, but the authors do not have a clear knowledge of the recent developments on thermochromism

Response:

Thank you for the comment.

To help further clarify the mechanism of thermochromism (with the help of Prof. Ge, our co-author, who is an expert on thermochromism), we have added text to describe how the color change occurs (SI, following Figure S14).

In addition, we have added six more references, beyond the 17 existing references on the topic of color change (including a 2018 review). We hope that these additions capture the recent developments and we remain open to include other recent developments.

We note that we are using well-established, commercial pigments and thus, the thermochromism is not the novel aspect of the manuscript. We simply use them because they change color with temperature.

REVIEWERS' COMMENTS:

Reviewer #1 (Remarks to the Author):

Thank you for addressing my concerns.

Reviewer #3 (Remarks to the Author):

The authors have addressed my comments